

# Cloud property datasets retrieved from AVHRR, MODIS, AATSR and MERIS in the framework of the Cloud_cci project

Martin Stengel[1], Stefan Stapelberg[1], Oliver Sus[1], Cornelia Schlundt[1], Caroline Poulsen[2],
Gareth Thomas[2], Matthew Christensen[2], Cintia Carbajal Henken[3], Rene Preusker[3], Jürgen Fischer[3],
Abhay Devasthale[4], Ulrika Willén[4], Karl-Göran Karlsson[4], Gregory R. McGarragh[5], Simon Proud[5],
Adam C. Povey[6], Roy G. Grainger[6], Jan Fokke Meirink[7], Artem Feofilov[8], Ralf Bennartz[9,10], Jedrzej
S. Bojanowski[11], and Rainer Hollmann[1]

[1]Deutscher Wetterdienst, Frankfurter Str. 135, 63067, Offenbach, Germany
[2]Rutherford Appleton Laboratory, Didcot, Oxfordshire, UK
[3]Institute for Space Sciences, Freie Universität Berlin, Carl-Heinrich-Becker-Weg 6-10, 12165 Berlin, Germany
[4]Swedish Meteorological and Hydrological Institute (SMHI), Norrköping, Sweden
[5]Department of Physics, University of Oxford, Clarendon Laboratory, Parks Road, Oxford OX1 3PU, UK
[6]National Centre for Earth Observation, University of Oxford, Oxford, OX1 3PU, UK
[7]Royal Netherlands Meteorological Institute (KNMI), De Bilt, The Netherlands
[8]Laboratoire de météorologie dynamique (LMD), Paris, France
[9]University of Wisconsin, Madison, Wisconsin, USA
[10]Vanderbilt University, Nashville, Tennessee, USA
[11]MeteoSwiss, Zurich, Switzerland

*Correspondence to:* Martin Stengel (martin.stengel@dwd.de)

**Abstract.** New cloud property datasets based on measurements from the passive imaging satellite sensors AVHRR, MODIS, ATSR2, AATSR and MERIS are presented. Two retrieval systems were developed that include components for cloud detection and cloud typing followed by cloud property retrievals based on the optimal estimation (OE) technique. The OE-based retrievals are applied to simultaneously retrieve cloud-top pressure, cloud particle effective radius and cloud optical thickness using

measurements at visible, near-infrared and thermal infrared wavelengths, which ensures spectral consistency. The retrieved cloud properties are further processed to derive cloud-top height, cloud-top temperature, cloud liquid water path, cloud ice water path and spectral cloud albedo. The Cloud_cci products are pixel-based retrievals, daily composites of those on a global equal-angle latitude-longitude grid, and monthly cloud properties such as averages, standard deviations and histograms, also on a global grid. All products include rigorous propagation of the retrieval and sampling uncertainties. Grouping the orbital

properties of the sensor families, six datasets have been defined, which are named: AVHRR-AM, AVHRR-PM, MODIS-Terra, MODIS-Aqua, ATSR2-AATSR and MERIS+AATSR, each comprising a specific subset of all available sensors. The individual characteristics of the datasets are presented together with a summary of the retrieval systems and measurement records on which the dataset generation were based. Example validation results are given, based on comparisons to well-established reference observations, which demonstrate the good quality of the data. Together with the ensured spectral consistency and

rigorous uncertainty propagation though all processing levels, the Cloud_cci datasets approach new benchmarks for climate data records of cloud properties based on passive imaging sensors.



For each dataset a Digital Object Identifier has been issued:

Cloud_cci AVHRR-AM: 10.5676/DWD/ESA_Cloud_cci/AVHRR-AM/V002

Cloud_cci AVHRR-PM: 10.5676/DWD/ESA_Cloud_cci/AVHRR-PM/V002

Cloud_cci MODIS-Terra: 10.5676/DWD/ESA_Cloud_cci/MODIS-Terra/V002

5    Cloud_cci MODIS-Aqua: 10.5676/DWD/ESA_Cloud_cci/MODIS-Aqua/V002

Cloud_cci ATSR2-AATSR: 10.5676/DWD/ESA_Cloud_cci/ATSR2-AATSR/V002

Cloud_cci MERIS+AATSR: 10.5676/DWD/ESA_Cloud_cci/MERIS+AATSR/V002

# 1   Introduction

Satellite-based datasets of geophysical variables evolve periodically by revisiting and maturing two essential aspects: the underlying radiance records and the retrieval systems applied. The first aspect is usually addressed by tracking down and collecting historic and new satellite recordings, by characterizing the accuracy and stability of the full measurement record,

and by applying new inter-calibration backwards through the entire record whenever new satellite missions provide better references. The second aspect is facilitated by: (1) continuously growing computer capabilities enabling the application of more advanced (and more computationally expensive) retrieval systems and the utilization of additional or more frequent auxiliary data which often undergo regular updates themselves, and (2) retrieval improvements which are often triggered when new satellite missions offer more accurate reference observations against which the retrieval systems can be validated.

In the last few decades, activities to process and reprocess global, high-quality cloud property datasets based on long-term satellite measurement records have been undertaken with increased effort. The backbone of most of the multi-decadal climate datasets of cloud properties has been the National Oceanic and Atmospheric Administration (NOAA) Polar Operational Environmental Satellites (POES) series. The Advanced Very High Resolution Radiometer (AVHRR) has been on board the NOAA satellites since the end of the 1970s (i.e. NOAA-5 and beyond). AVHRR is a passive imaging sensor, where the source

of measured radiation is not emitted by the instrument. Instead, the upwards reflected solar and emitted thermal radiation is measured at the top of the atmosphere (TOA). This is done in abutting pixels that assemble a seamless image. With its four to six spectral channels, AVHRR allows the retrieval of key cloud properties. AVHRR has been a significant contributor to many global cloud climatologies, e.g. the International Satellite Cloud Climatology Project (ISCCP, Schiffer and Rossow, 1983; Rossow and Schiffer, 1999), the Pathfinder extended dataset (PATMOS-x, Heidinger et al., 2014) and the Climate Monitoring

Satellite Application Facility's (CM SAF) cloud, albedo and radiation dataset (CLARA-A1/A2, Karlsson et al., 2013, 2016).

Since the 1990s the National Aeronautics and Space Administration (NASA) and the European Space Agency (ESA) have launched research satellite missions, e.g. Terra, Aqua, the European Remote-sensing Satellite (ERS-1/2) and the Environmental Satellite (Envisat), that carry AVHRR heritage sensors. These are the Moderate-resolution Imaging Spectroradiometer (MODIS), the Along-Track Scanning Radiometers (ATSR-1/2) and the Advanced Along-Track Scanning Radiome-

ter (AATSR), which provide an increased number of spectral channels as well as higher spatial resolution ($\leq$1km footprint size) than AVHRR. The cloud datasets derived from these measurement records cover more than one decade and are thus becoming



useful for climate studies. Examples of related cloud property datasets are the Global Retrieval of ATSR cloud parameters and evaluation (GRAPE, Sayer et al., 2011) for ATSR/AATSR, the NASA MODIS Collection 5 (Platnick et al., 2003; King et al., 2003) and Collection 6 (Baum et al., 2012; Platnick et al., 2014; Marchant et al., 2016; Platnick et al., 2017). The MODIS and
ATSR/AATSR sensors include the spectral channels of AVHRR, but have additional ones in the visible, near-infrared and, in case of MODIS, also in the thermal infrared. However, even when restricted to the AVHRR heritage channels, their increased spatial resolution as well as their contribution to increasing the observation frequency motivate their consideration in climate research, in particular in conjunction with AVHRR.

Most of the aforementioned cloud property datasets have improved over the years and have now reached quality levels that facilitate qualitative and quantitative assessments of clouds in the Earth's climate system (e.g. Norris et al., 2016; Sun et al., 2015; Enriquez-Alonso et al., 2015; Carro-Calvo et al., 2016; Terai et al., 2016) including studies to understand cloud processes and the evaluation of atmospheric models. However, there is still potential for advancing such datasets.

A common shortcoming of existing datasets is the absence of uncertainty information for pixel-level retrievals (Level-2 data) as well as for daily and monthly averages (Level-3 data). These uncertainties should be derived using a mathematically sound framework with uncertainty propagation. Another improvement to cloud property datasets is to ensure that the properties retrieved using mainly shortwave measurements are radiatively consistent with those mainly based on thermal infrared measurements. This is known as spectral consistency and is important to ensure that subsequent simulations of TOA radi-
ances using these retrieved cloud properties match the measured radiances in all spectral bands. The same can be inferred for TOA broad-band fluxes produced using the retrieved parameters. Spectral consistency is not maintained in existing cloud retrievals (e.g. Ham et al., 2009) despite being of particular importance to, for example, studies investigating the impact of cloud properties and their change on TOA broadband fluxes and latent heating rates.

The ESA Cloud_cci project covers the cloud component within ESA's Climate Change Initiative (Hollmann et al., 2013).
The overarching aim of the ESA Cloud_cci project has been the generation of state-of-the-art cloud property datasets based on European and non-European satellite missions including the investigation of their synergistic capabilities. This was achieved by:

- Characterizing and advancing measurement records of passive sensors of ESA and non-ESA satellite missions (Karlsson and Johansson, 2014);

- Developing physical retrieval systems for cloud properties with spectral consistency over all utilized spectral bands;

- Generating multi-decadal global cloud datasets, based on both single sensors and on a synergistic use of multiple sensors, including uncertainty estimates which are propagated through all processing levels.

The retrieval systems presented in this paper are based on the optimal estimation (OE) technique (e.g., Rodgers, 2000) and are used to derive a set of cloud variables simultaneously using the visible, near-infrared and thermal infrared measurements.
The retrieval systems were used to generate cloud property datasets spanning the entire available measurement record from 1982 until 2014. In the first phase of Cloud_cci project, prototype versions of the datasets (version 1.0) were generated. In



this paper, the version 2.0 of the Cloud_cci datasets are introduced by presenting a concise overview of the most important technical and scientific aspects. Section 2 gives an overview of the Cloud_cci datasets. This includes a description of the underlying measurement records, the retrieval systems used, the cloud variables produced at different processing levels, and

the propagation of the Level-2 uncertainties. In Section 3 selected examples of the datasets are shown and discussed, and Section 4 reports the most important validation results. Section 5 summarizes the paper.

## 2   Composition of the Cloud_cci datasets

The following satellites and sensors were used in Cloud_cci:

- AVHRR on board the NOAA POES satellites (NOAA-7,-9,-11,-12,-14,-15,-16,-17,-18,-19) and on board the European Organisation for the Exploitation of Meteorological Satellites (EUMETSAT) Meteorological operational satellite Metop-A;

- MODIS on board NASA's Aqua and Terra satellites;

- ATSR-2 and AATSR on board ESA's research satellites ERS-2 and Envisat;

- The Medium Resolution Imaging Spectrometer (MERIS) also on board Envisat

Considering imaging and orbital characteristics of the sensors processed, six datasets were compiled as given in Table 1. For

10   all datasets Digital Object Identifiers (DOIs) have been established (also given in Table 1). Figure 1 reports the local equator crossing times of all sensors considered.

**Table 1.** List of Cloud_cci datasets together with the corresponding retrieval scheme, the sensor(s), satellite(s) used and the time period covered as well as the Digital Object Identifiers (DOIs) issued.

| Cloud_cci dataset | Retrieval used | Sensor(s) | Satellite(s) | Temporal coverage | DOI |
|---|---|---|---|---|---|
| AVHRR-PM | CC4CL | AVHRR | N-7,9,11,14,16,18,19 | 1982-2014 | 10.5676/DWD/ESA_Cloud_cci/AVHRR-PM/V002 |
| AVHRR-AM | CC4CL | AVHRR | N-12,15,17,Metop-A | 1991-2014 | 10.5676/DWD/ESA_Cloud_cci/AVHRR-AM/V002 |
| MODIS-Aqua | CC4CL | MODIS | Aqua | 2002-2014 | 10.5676/DWD/ESA_Cloud_cci/MODIS-Aqua/V002 |
| MODIS-Terra | CC4CL | MODIS | Terra | 2000-2014 | 10.5676/DWD/ESA_Cloud_cci/MODIS-Terra/V002 |
| ATSR2/AATSR | CC4CL | ATSR2,AATSR | ERS2,Envisat | 1995-2012 | 10.5676/DWD/ESA_Cloud_cci/ATSR2-AATSR/V002 |
| MERIS-AATSR | FAME-C | MERIS,AATSR | Envisat | 2003-2011 | 10.5676/DWD/ESA_Cloud_cci/MERIS-AATSR/V002 |

AM: ante meridiem, before noon; PM: post meridiem, after noon; CC4CL: Community Cloud retrieval for CLimate, FAME-C: Freie Universität Berlin AATSR MERIS Cloud retrieval; N: NOAA



Earth System
Science
Data

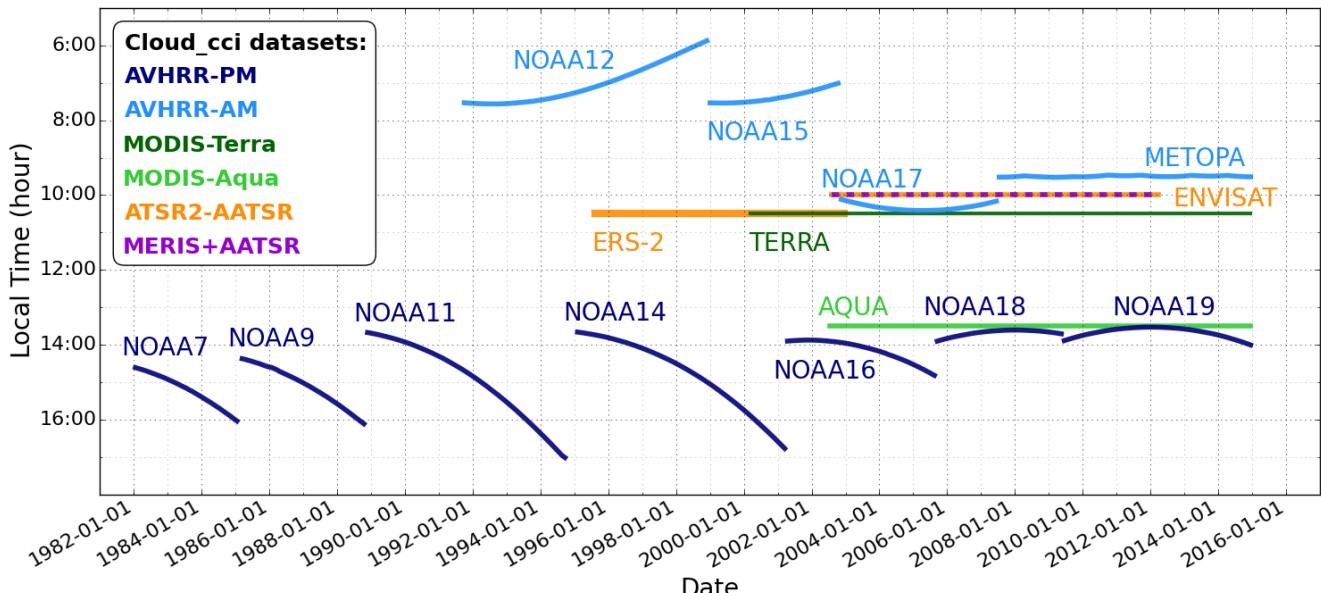

**Figure 1.** Overview of all sensors processed in Cloud_cci and their duration as a function of the daytime equator crossing time (AM: ante meridiem, before noon; PM: post meridiem, after noon). Sensors belonging to the same dataset are shown in the same color.

## 2.1 The measurement records used

Measurements from the passive imaging sensors AVHRR, MODIS, ATSR2, AATSR and MERIS sensors were used to produce the Cloud_cci datasets. Each sensor has different imaging characteristics, such as differences in swath width, which leads to varying observation frequency for any given position on Earth. All sensors operate in a sun-synchronous polar orbit. Each individual orbit has a ascending and descending part, which roughly corresponds to either daytime or night-time conditions, thus are also referred to as daytime and night-time node). For the MERIS+AATSR dataset, the night-time observations are ignored. Individual AVHRR and MODIS sensors cover the globe nearly twice a day with the daytime and night-time nodes
5   of their orbits. Due to their narrow swath width, ATSR2 and AATSR need three days to cover the full globe with daytime and night-time observations (Figure 2). With respect to the local equator crossing time of the daytime node, the AVHRR-carrying satellites were separated into AM (a.m. - ante meridiem, before noon) and PM (p.m. - post meridiem, after noon). In the following sections further characteristics of the measurement data that form the basis of the Cloud_cci datasets are summarized.

10  ### 2.1.1 AVHRR

The second and third generation of the AVHRR sensor (AVHRR/2 and AVHRR/3) provide measurements in two visible, one near-infrared and two thermal infrared channels with the following (approximate) central wavelengths: 0.6µm, 0.8µm,



a) AVHRR-PM/NOAA-18    b) AVHRR-AM/Metop-A    c) MODIS-Aqua

d) MODIS-Terra    e) ATSR2-AATSR/Envisat    f) MERIS+AATSR/Envisat

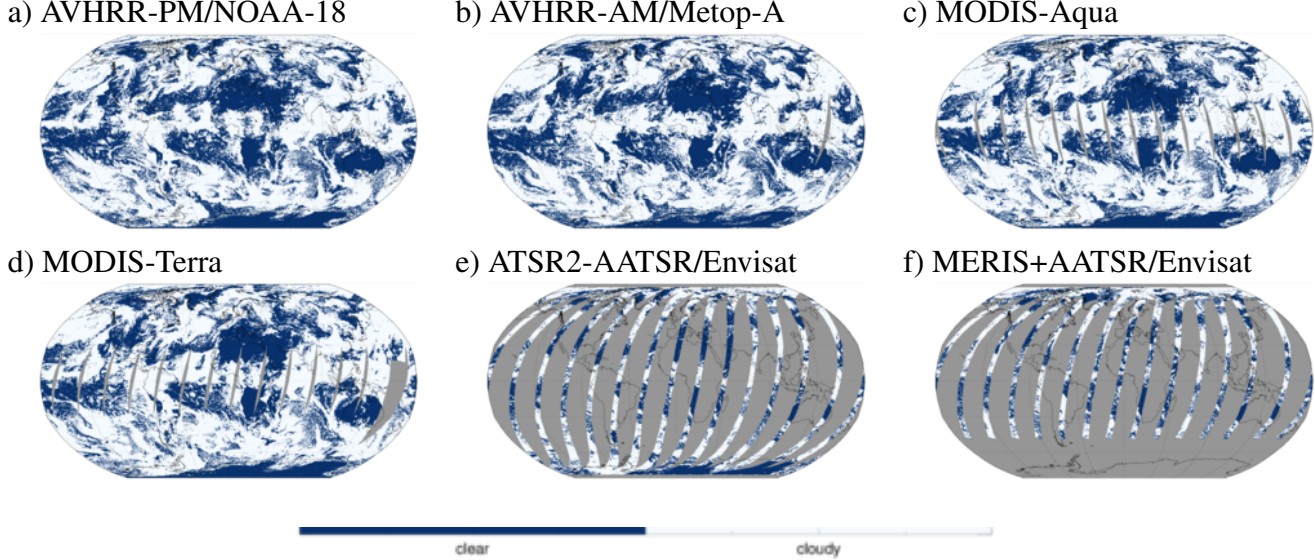

**Figure 2.** Daily Level-3U cloud mask composites for 2008/06/22 demonstrating the spatial coverage of the daytime nodes of selected sensors within 24 hours. See Table 3 for definition of Level-3U.

3.7μm, 10.8μm, 12.0μm. Exceptions are daytime observations of NOAA-16 in 2001-2003 and the entire records of NOAA-17 and Metop-A, for which a 1.6μm channel was switched on during the day and used instead of the 3.7μm channel. The AVHRR swath width is 2399km, which facilitates full global coverage (daytime and night-time) twice daily. More information about the AVHRR sensor can for example be found at https://www.wmo-sat.info/oscar/instruments/view/62. The Cloud_cci AVHRR-AM and AVHRR-PM datasets are based on measurements from AVHRR/2 and AVHRR/3 on board prime polar-orbiting NOAA POES and Metop satellites; AVHRR-PM: NOAA-7, NOAA-9, NOAA-11, NOAA-14, NOAA-16, NOAA-18 and NOAA-19; and AVHRR-AM: NOAA-12, NOAA-15, NOAA-17 and Metop-A. All measurements used are Global Area Coverage (GAC) data with a footprint size of 1.1km x 4.4km and a sampling distance of 3.3km along track and 5.5km across track between the centres of the GAC footprints. GAC data is derived from the originally measured Local Area Coverage (LAC, footprint size: 1.1km x 1.1km) data by on-board averaging of four neighbouring LAC pixels every third scan line. Only the GAC record is available for AVHRR with global and nearly seamless temporal coverage since the early NOAA satellites. The two visible channels and the 1.6μm channel (if present) were intercalibrated using MODIS Collection 6 measurements (Heidinger et al.; publication currently in preparation, an update of Heidinger et al. (2010)). For the IR channels no further calibration was performed beyond the on-board blackbody calibration.

### 2.1.2 MODIS

MODIS is a 36-channel passive imaging sensor with a swath width of 2330km. More information on the MODIS sensor is available at https://www.wmo-sat.info/oscar/instruments/view/296. Out of the original 36 spectral channels, measurements





from 5 of them (the AVHRR heritage channels, MODIS channel numbers 1,2,20,31,32) were used to retrieve cloud properties
from MODIS-Terra (in 2000-2014) and MODIS-Aqua (in 2002-2014). The MODIS sensor is calibrated on board. The calibration approach employs radiometric, spatial, and spectral calibrators and the moon as reference (Xiong and Barnes, 2006). MODIS sensors are well-known for their high calibration accuracy. MODIS Level-1b data of the Collection 6 release were obtained from NASA. The Collection 6 Level-1 data are expected to show several improvements over Collection 5 data due to improved calibration methodologies (see for example http://mcst.gsfc.nasa.gov/calibration/collection_6_info). The footprint size of the MODIS Level-1 data is 0.25km x 0.25km to 1km x 1km (depending on channel), here we used data at 1km x 1km resolution for all channels.

### 2.1.3 ATSR2 and AATSR

The passive imaging sensors ATSR2 and AATSR have seven spectral channels in the solar, near-infrared and thermal infrared range with central wavelengths between 0.55μm and 12μm, of which five (the AVHRR heritage channels; ATSR2/AATSR channel numbers: 2, 3, 5, 6, 7) were used. At nadir the ATSR2 and AATSR pixel resolution is approximately 1km x 1km with a swath width of 500km. The sensor is designed to be self-calibrating. Two integrated thermally controlled black-body targets are used for calibration of the thermal channels, while an opal visible calibration target illuminated by sunlight is used for calibration of visible and near-infrared channels. More information about the ATSR-2 and AATSR sensors is available at https://www.wmo-sat.info/oscar/instruments/view/56 and https://www.wmo-sat.info/oscar/instruments/view/2. The version 3.1 of the ATSR2 and AATSR Level-1 data was used. The available ATSR2 Level-1 record covers 1995-2002, while the AATSR Level-1 record covers 2002-2012.

### 2.1.4 MERIS

MERIS is a passive imaging sensor whose measurements were synergistically combined with AATSR measurements in this study, making use of the fact that both sensors are mounted on the same platform (Envisat) but have complementary spectral characteristics. The pixels of both sensors were spatially matched. MERIS measurements outside the AATSR swath width (500km, which is narrower than the MERIS swath width of 1150km), were not used. The collocated synergy product has a swath width of 493 pixels. This is related to collocating the curved AATSR grid with the MERIS grid. More information on the matching procedure can be found in Gómez-Chova et al. (2009). The spatial resolution of the MERIS reduced resolution mode is 1.2km x 1km, thus very similar to AATSR. More information about the MERIS sensor is available at https://www.wmo-sat. info/oscar/instruments/view/277. MERIS Level-1 data of the third reprocessing has been used (https://earth.esa.int/documents/ 700255/707222/A879-NT-017-ACR_v1.0.pdf/6fa86bec-9945-4e39-808e-3801f2e3962b). In addition to the above-mentioned AVHRR heritage channels of AATSR, MERIS channel 11 (spectrally located in the Oxygen-A absorption band around 761nm) and channel 10 (a nearby window channel located around 753nm) were used. An empirical stray light correction was applied to the reflectance of the MERIS Oxygen-A absorption band channel (Lindstrot et al., 2010). For this correction, the spectral smile effect in the MERIS measurements (Bourg et al., 2008), which is the variation of the channel center wavelength along the field of view, as well as the amount of stray light in the MERIS Oxygen-A absorption band channel, was determined.



## 2.2 Retrieval systems

Based on a comprehensive comparison to existing cloud property retrieval systems applicable to passive imaging sensors (Stengel et al., 2015), the two Cloud_cci algorithms were further developed and then used to generate the Cloud_cci climate records. For datasets derived from AVHRR and from the AVHRR heritage channels of MODIS, ATSR-2 and AATSR, the

30 Community Cloud retrieval for CLimate (CC4CL; Sus et al., 2017; McGarragh et al., 2017b) retrieval system was employed. For the MERIS+AATSR dataset, the Freie Universität Berlin AATSR MERIS Cloud (FAME-C; Carbajal Henken et al., 2014) retrieval system was employed. Common to both systems is the OE technique, which uses a Levenburg-Marquardt (Levenberg, 1944; Marquardt, 1963; Rodgers, 2000) non-linear inversion method to iteratively fit simulated TOA radiances to the measured TOA radiances. The ability to include a prior knowledge of the retrieved quantities is built into the method. The OE technique supports comprehensive error propagation, allowing measurement error, forward model error (due to approximations and assumptions, which are made in the modelling of TOA radiance) and uncertainties in a priori knowledge to be combined to give

a rigorous estimate of the uncertainty on retrieved values on a pixel-by-pixel basis.

CC4CL and FAME-C were employed to primarily retrieve the following cloud properties: cloud mask (CMA), cloud phase (CPH), cloud optical thickness (COT), cloud effective radius (CER) and cloud-top pressure (CTP). From these properties, cloud-top temperature (CTT), cloud-top height (CTH), liquid water path (LWP), ice water path (IWP) and cloud albedo (CLA) were also determined. A short description of all cloud properties is given in Table 2. The next two subsections briefly summarize

the main characteristics of CC4CL and FAME-C with in-depth details to be found in the references given therein.

### 2.2.1 CC4CL

The CC4CL retrieval system has three main components: cloud detection to retrieve CMA, cloud typing to retrieve CPH and an OE retrieval of COT, CER and CTP. The three components are framed by a pre-processing (e.g. spatio-temporal mapping of all data fields and clear-sky radiative transfer simulations) and a post-processing (e.g. merging, consistency check, quality

control). The cloud detection is performed by applying an artificial neural network (ANN) that uses the AVHRR heritage channel measurements, illumination, scan angles, and auxiliary data as input. The ANN was trained to mimic the COT of the Cloud-Aerosol Lidar with Orthogonal Polarization (CALIOP, Winker et al., 2009), which is the main payload of the Cloud-Aerosol Lidar and Infrared Pathfinder Satellite Observations (CALIPSO) satellite. Cloudy pixels are identified where the ANN-estimated COT exceeds pre-defined thresholds (with the remaining pixels classified as clear), thus defining the CMA.

Based on CALIOP data in the ANN training set, a cloud mask uncertainty is determined. The cloud typing is based on a threshold decision tree documented in Pavolonis and Heidinger (2004) and Pavolonis et al. (2005). Various cloud types exist for either liquid or ice phase, which allows the simplification to a binary CPH information. The central part of CC4CL is an OE estimation of COT, CER and CTP, which is based on earlier developments of the Oxford RAL retrieval of Aerosol and Cloud retrieval (ORAC, Poulsen et al., 2012; Watts et al., 1998), but has undergone major updates since then. As mentioned

above, a cloud model is iteratively modified to fit the simulated radiances to the measurements. For this, a fast forward radiative transfer model is included using scalar reflectance, transmission and emission operators. These operators interact with (1) the



direct beam solar source and/or the diffuse thermal source from above and (2) both direct and diffuse surface reflectance from a bidirectional reflectance distribution function (BRDF) surface in the solar channels, and diffuse atmospheric and surface emission in the thermal channels from below. The operators are a function of the state vector elements COT and CER, and

solar and instrument geometry and compiled in Look-Up Tables (LUTs) precalculated using the DIScrete Ordinates Radiative Transfer (DISORT, Stamnes et al., 1988) solver. The simulations were done separately for liquid and ice clouds. Liquid clouds are represented with a modified gamma distribution (Hansen and Travis, 1974) and ice cloud single scattering properties are taken from Baran et al. (2005).

Clear-sky tranmittance and radiance profiles are computed using the Radiative Transfer for Television Infrared Observation Satellite Program (TIROS) Operational Vertical Sounder (TOVS) (RTTOV, Eyre, 1991; Saunders et al., 1999) model version 11.3 (Hocking et al., 2014). For each iteration the above-cloud and below-cloud clear-sky transmittances and radiances are interpolated from the corresponding RTTOV profiles as a function of CTP. From the derived state vector variables, the prop-

erties CTT, CTH, CLA, LWP and IWP are inferred with LWP and IWP calculations being based on Stephens (1978) for all cloudy conditions. A full list of retrieved cloud variables is given in Table 2. The reader is referred to Sus et al. (2017) and McGarragh et al. (2017b) for more details on CC4CL. The CC4CL system was used for the generation of the Cloud_cci datasets AVHRR-PM, AVHRR-AM, MODIS-Aqua, MODIS-Terra and ATSR2-AATSR.

### 2.2.2 FAME-C

FAME-C is an OE-based retrieval system for cloud properties using TOA radiance measurements from AATSR and MERIS. As an initial step a cloud detection is performed as described in Hollstein et al. (2015). This is followed by the cloud typing procedure of Pavolonis et al. (2005) and Pavolonis and Heidinger (2004), which is additionally simplified to provide binary liquid-ice information. For daytime pixels identified as cloudy and assigned with a cloud type, an OE-based retrieval of COT and CER is performed. The OE retrieval was initially based on developments documented in Walther and Heidinger (2012).

Required LUTs for mapping cloud properties to visible and near-infrared reflectances were composed through radiative transfer simulations utilizing the Matrix Operator Model (MOMO Fischer and Grassl, 1984; Fell and Fischer, 2001). From the retrieved COT and CER, LWP and IWP are computed using Stephens (1978) for liquid clouds and optically thick ice clouds. For optically thin ice clouds, the conversion of Heymsfield et al. (2003) is applied. Using the retrieved CPH and COT, a cloud-top temperature retrieval is conducted using the AATSR thermal infrared channels. Radiative transfer simulations for AATSR are done using

RTTOV version 11.2. The CTT is further converted to CTH and CTP using collocated NWP profiles of pressure, temperature and height. These CTP values are provided as first guess to a second CTP retrieval based on MERIS measurements in an Oxygen-A absorption band channel and a nearby window channel. The difference in sensitivity of both cloud height retrievals to different kinds of cloudy situations was analyzed in Carbajal Henken et al. (2015).

The full list of retrieved cloud properties using the FAME-C system largely overlaps with the CC4CL output, and is thus

also given in Table 2. The reader is referred to Carbajal Henken et al. (2014) for more details on the FAME-C. The FAME-C system was used for the generation of the Cloud_cci MERIS+AATSR dataset.



## 2.3 Product definitions

The full suite of Level-2 cloud properties derived from both retrieval systems is: CMA, CPH, CTP, CTH, CTT, COT, CER, CWP, and CLA (at two wave lengths), where CWP represents either LWP in pixels with liquid clouds or IWP in pixels with ice clouds. Nearly all of these properties are accompanied by uncertainty measures that are direct outputs of OE (or derived from them). Exceptions are CC4CL CMA, for which the uncertainty is empirically estimated from validation work (Sus et al., 2017). Furthermore, CMA from FAME-C and CPH from both retrieval systems are not accompanied by uncertainty information yet. Besides Level-2, two additional processing levels exist: Level-3U and Level-3C, which are explained in Section 2.4.

**Table 2.** List of generated cloud properties. See Section 2.4 for more information on the processing levels Level-2, Level-3U and Level-3C.

| Variable | Abbreviation | Definition |
| --- | --- | --- |
| Cloud mask | CMA | A binary cloud mask per pixel (Level-2, Level-3U) |
| Cloud fractional cover | CFC | Subsequently calculated monthly total cloud fractional cover (Level-3C); also separated into 3 vertical classes (high, mid-level, low clouds) following ISCCP classification of Rossow and Schiffer (1999). |
| Cloud phase | CPH | The thermodynamic phase of the cloud (binary: liquid or ice; in Level-2, Level-3U) |
| Liquid cloud fraction | | The monthly liquid cloud fraction (Level-3C) using the binary cloud phase information. |
| Cloud optical thickness | COT | The line integral of the absorption and the scattering coefficients along the vertical in cloudy pixels |
| Cloud effective radius | CER | The projected-area-weighted mean radius of the cloud drop and crystal particles, respectively. |
| Cloud-top pressure | CTP | The air pressure at the top of the uppermost cloud layer - direct output of OE. |
| Cloud-top height | CTH | Height of cloud top, inferred from CTP using ERA-Interim (Dee et al., 2011) profiles. |
| Cloud-top temperature | CTT | Air temperature at the cloud top, inferred from CTP using ERA-Interim profiles. |
| Cloud water path (containing ice and liquid water path) | CWP (LWP, IWP) | The vertically integrated liquid/ice water content in a cloudy column; derived from CER and COT following Stephens (1978) |
| Joint cloud property histogram | JCH | A spatially resolved two-dimensional histogram of combinations of COT and CTP for each spatial grid cell (Level-3C only) |
| Spectral cloud albedo at 0.6μm | CLA_vis006 | The black-sky albedo derived for channel 1 (0.67μm) and 2 (0.87μm*), respectively |
| Spectral cloud albedo at 0.8μm | CLA_vis008* | (experimental product) |

* For FAME-C, the cloud albedo is derived at 1.6μm instead of 0.87μm.

## 2.4 From pixel-based retrieval data (Level-2) to daily and monthly properties (Level-3)

Level-2 data were the input to the Level-3 processing and underwent a spatio-temporal sampling and averaging. Level-3U products are global composites, defined on a latitude-longitude grid at 0.05° x 0.05° resolution. Level-3U fields hold Level-2 data which were sampled into the Level-3U grid within a 24-hour time window. The most important aspects of this sampling



**Table 3.** Processing levels of Cloud_cci data products. The footprint refers to the area on the Earth's surface that is covered by one sensor pixel at nadir view.

| Processing level | Footprint size | Description |
|---|---|---|
| Level-2 (Pixel data) | MODIS: 1km AATSR: 1km AVHRR: 5km MERIS+AATSR: 1km | Retrieved cloud properties at the same resolution and location as the native sensor measurement (Level-1) |
| Level-3U (Daily composites) | Global equal-angle, latitude-longitude grid with 0.05° resolution (MODIS-Europe: 0.02°) | Cloud properties of Level-2 data granules sampled to a global grid without combining any observations from overlapping orbits. Only sampling is done. Common alternative notations for this processing level are Level-2B or Level-2G. Temporal coverage of this product is one day. |
| Level-3C (Monthly averages & histograms) | Global equal-angle, latitude-longitude grid with 0.5° resolution | Cloud properties combined (averaged) from a single sensor into a global grid; sampled for the histograms. Temporal coverage of this product is one month. |

**Table 4.** Cloud_cci product portfolio, also featuring day/night and liquid/ice separation for some properties. All products listed exist for each dataset.

|  | Level-2 pixel level | Level-3U daily composites | Level-3C monthly averages | Level-3C monthly histograms |
|---|---|---|---|---|
| CMA/CFC | ✓ | ✓ | ✓ (day/night high/mid/low) | - |
| CTP, CTH, CTT | ✓ | ✓ | ✓ | ✓ (liquid/ice) |
| CPH | ✓ | ✓ | ✓ (day/night) | - |
| COT | ✓ | ✓ | ✓ (liquid/ice) | ✓ (liquid/ice) |
| CER | ✓ | ✓ | ✓ (liquid/ice) | ✓ (liquid/ice) |
| LWP | ✓ (as CWP) | ✓ (as CWP) | ✓ | ✓ |
| IWP | ✓ (as CWP) | ✓ (as CWP) | ✓ | ✓ |
| CLA | ✓ (0.6/0.8μm*) | ✓ (0.6/0.8μm*) | ✓ (0.6/0.8μm*) | ✓ (liquid/ice) |
| JCH | N/A | N/A | N/A | ✓ (liquid/ice) |

\* For FAME-C, the cloud albedo is derived at 1.6μm instead of 0.87μm.

5  procedure are: (1) only that Level-2 pixel that has the lowest satellite zenith angle is kept in each Level-3U grid cell and (2) the actual footprint size of each pixel is considered (which depends on the sensor and scan angle), which can lead to more than one Level-3U grid cell being filled by one single Level-2 pixel observations. The Level-3U composition was done for each day; keeping the ascending and descending nodes of the orbits in separate fields, which roughly corresponds to separating



**Table 5.** Bin borders of Cloud_cci Level-3C histograms of cloud-top pressure (CTP), cloud-top temperature (CTT), cloud optical thickness (COT), cloud effective radius (CER), liquid and ice water path (CWP), cloud albedo (CLA), and Joint Cloud property Histogram (JCH).

| Cloud_cci variable | Bin borders in Level-3C monthly histograms |
|---|---|
| CTP [hPa] | 1, 90, 180, 245, 310, 375, 440, 500, 560, 620, 680, 740, 800, 875, 950, 1100 |
| CTT [K] | 200, 210, 220, 230, 235, 240, 245, 250, 255, 260, 265, 270, 280, 290, 300, 310, 350 |
| COT | 0, 0.3, 0.6, 1.3, 2.2, 3.6, 5.8, 9.4, 15, 23, 41, 60, 80, 99.99, 1000 |
| CER [µm] | 0, 3, 6, 9, 12, 15, 20, 25, 30, 40, 60, 80 |
| CWP [$\mathrm{g\,m^{-2}}$] | 0, 5, 10, 20, 35, 50, 75, 100, 150, 200, 300, 500, 1000, 2000, 100000 |
| CLA_vis006/008* | 0, 0.1, 0.2, 0.3, 0.4, 0.5, 0.55, 0.6, 0.65, 0.7, 0.75, 0.8, 0.9, 1 |
| JCH | See COT and CTP bins |

\* For FAME-C, the cloud albedo is derived at 1.6µm instead of 0.87µm.

daytime and night-time observations. The Level-3U products also hold a variety of ancillary data information apart from the
10 retrieved cloud properties. Taking advantage of the high spatial resolution of the MODIS sensor, additional Level-3U products
were produced for MODIS for a 0.02° x 0.02° grid covering the European area within 15°W to 45°E and 35°N to 75°N (not
shown).

Level-3C products are defined on a latitude-longitude grid with 0.5° x 0.5° resolution and hold monthly summaries of the
Level-2 data, such as averages and standard deviation. In addition, monthly histograms were composed for CTP, CTT, CER,
COT, CWP, CLA, each separated into liquid and ice clouds, and for combinations of COT and CTP (Joint Cloud property
Histogram, JCH). Table 3 summarizes most important characteristics of all processing levels. The binning of the Level-3C
histograms is given in Table 5. Table 4 reflects the available cloud properties for each processing level.

Due to differences in spatial resolution and swath width between the considered sensors, the spatio-temporal observation fre-
quency is very different. The effect of this on monthly scale is demonstrated in Figure 3 by the number of daytime observations
per 0.5° grid cell per month.

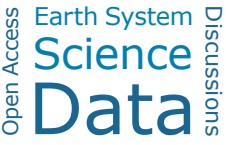

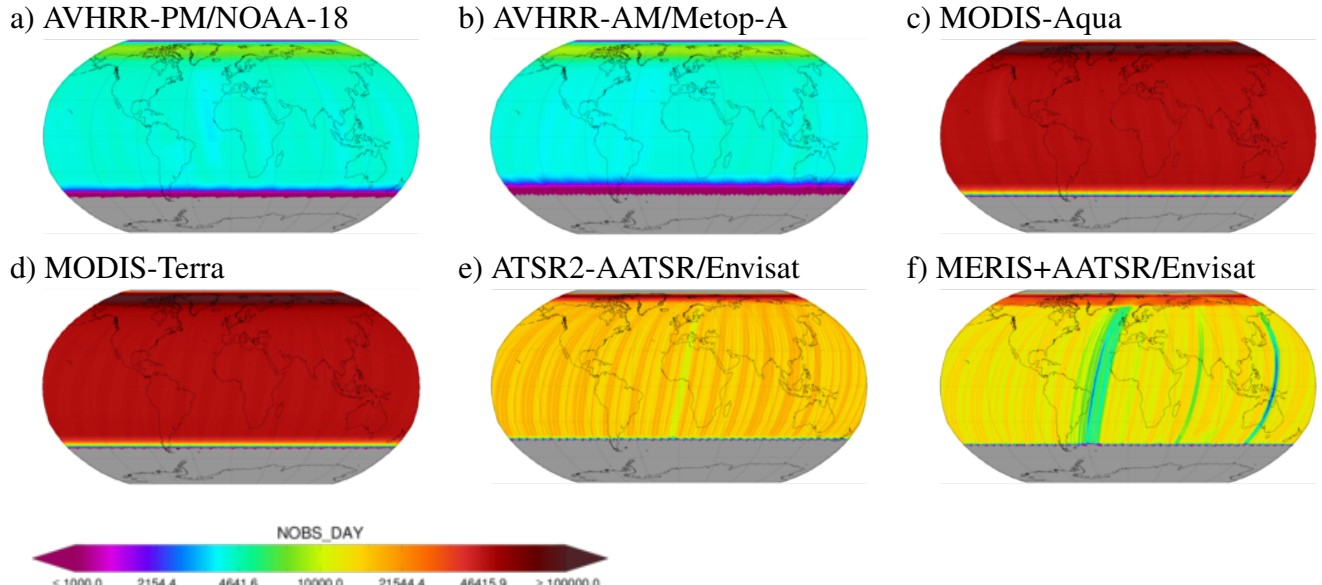

**Figure 3.** Number of daytime observations (pixels) per 0.5° grid cell for June 2008 and (a) AVHRR-PM/NOAA-18, (b) AVHRR-AM/Metop-A, (c) MODIS-Aqua, (d) MODIS-Terra, (e) AATSR and (f) MERIS+AATSR. For the sake of comparability only the daytime number is shown because no night-time observations are included in MERIS+AATSR. Gray-shaded areas indicate regions with no day-light, thus without daytime observations.

### 2.4.1 Propagating the uncertainties

Different metrics can be used to represent the uncertainty of monthly mean Level-3C products. A simple and often used metric is the standard deviation $\sigma_{\text{std}}$ (Equation 1) calculated over the same set of retrieved pixels ($x_i$) that is used for the calculation of the mean ($\langle x \rangle$):

$$\sigma_{\text{std}}^2 = \frac{1}{N} \sum_{i=1}^{N} (x_i - \langle x \rangle)^2 \tag{1}$$

with N being the number of pixels.

The OE approach provides pixel-based retrieval uncertainties ($\sigma_i$) that are based on mathematically consistent propagation of the uncertainties of the input data (e.g. auxiliary data, measurement data, background errors) into the Level-2 product space (see for example McGarragh et al. (2017b) and Sus et al. (2017)). For the Cloud_cci datasets, the Level-2 uncertainties were further propagated into Level-3C products by two measures: the mean of the pixel-based uncertainties ($\langle \sigma_i \rangle$, Equation 2) and the mean of the squares of the pixel-based uncertainties ($\langle \sigma_i^2 \rangle$, Equation 3).

$$\langle \sigma_i \rangle = \frac{1}{N} \sum_{i=1}^{N} (\sigma_i) \tag{2}$$





$$\langle \sigma_i^2 \rangle = \frac{1}{N} \sum_{i=1}^{N} (\sigma_i^2) \tag{3}$$

With these measures it is possible to include the OE-based Level-2 uncertainties when quantifying both the true, natural variability ($\sigma_\text{true}$) of the observed geophysical variable (Equation 4) and the uncertainty of the calculated mean ($\sigma_{\langle x \rangle}$, Equation 5).

$$\sigma_\text{true}^2 = \sigma_\text{std}^2 - (1-c)\langle \sigma_i^2 \rangle \tag{4}$$

$$\sigma_{\langle x \rangle}^2 = \frac{1}{N}\sigma_\text{true}^2 + c\langle \sigma_i \rangle^2 + (1-c)\frac{1}{N}\langle \sigma_i^2 \rangle \tag{5}$$

These equations assume a bias-free Gaussian distribution for both the Level-2 uncertainties and the retrieved variables. This assumption is inaccurate for many variables of the presented properties, which introduces some limitations to the presented approach. Hence, the propagated uncertainties are meant to be experimental for these dataset versions. Assuming Gaussian and bias-free distributions, the estimated natural variability represents the standard deviation (thus the inner 68% percentile) of the distribution around the true mean. Furthermore, the estimated uncertainty of the monthly mean represents the 68% confidence interval around the calculated monthly mean.

The given framework was applied to all cloud properties and their OE-based uncertainties in the generation of the Level-3C products. The results are discussed using the example of COT from the AVHRR-PM dataset for June 2008. Figure 4 shows global maps of monthly mean COT and the corresponding monthly standard deviation (panels a and b) as calculated from the retrieved Level-2 values. The estimate of the true variability is shown along with the estimated uncertainty of the calculated mean (panel c and d) for an uncertainty correlation ($c$) of 0.1. Panels e and f show the results for an uncertainty correlation of 1.0. The exact correlation is not known and it is likely to have spatial and temporal variability. Due to this, two very different values have been chosen to illustrate the sensitivity. In this example, regions with high mean COT tend to be characterized by high spatio-temporal variability in the underlying data, which is apparent in the increased standard deviation. An exception is the northern part of the Atlantic and Pacific Oceans, where the standard deviation is low while the mean COT is high. A noticeable feature is found in the stratocumulus regions, which are located in the eastern parts of the subtropical ocean regions. There, the COT is very stable, thus shows low standard deviations. Another very dominant feature is a band of rather high mean COT accompanied with high variability, in the storm track regions of the Southern Oceans.

Now, assuming a rather low uncertainty correlation of 0.1, the estimated true variability becomes very small (Figure 4c). This is due to the second term on the right side of Equation 4, in which the mean of the squared retrieval uncertainties is not significantly reduced when using a correlation of 0.1. In other words, if the retrieval uncertainties are only slightly correlated, they contribute to a broadening of the observed COT distribution, causing only a minor systematic shift of the distribution. In this scenario, the retrieval uncertainties can explain a large portion of the observed standard deviation. In some regions



the second term of Equation 4 is even larger than the first term (the observed variance), which results in negative values of the estimated natural variability. Such negative values are non-physical and indicate an improper correlation in corresponding

regions. They have been set to 0 in the plots. The uncertainty of the mean is also relatively small for a correlation of 0.1 (Figure 4d). This is due to all three terms of Equation 5 becoming small: the third term because of the relatively large N, the second term because of the low correlation and the first term because of the low estimated natural variability, additionally minimized by the division by N. In other words, having small systematic uncertainties (i.e. a low uncertainty correlation) leads to a low uncertainty of the mean, which is dominated by sampling uncertainties decreasing with increasing N.

As a second example, an uncertainty correlation of 1.0 (panels e and f of Figure 4) is considered. The second term on the right hand side of Equation 4 vanishes in this case, leading to the estimated true variability being equal to the observed standard deviation. The uncertainty of the mean is also increased, which is in contrast to the previous scenario with low correlation. A correlation of 1.0 eliminates the third term of Equation 5. The first term decreases with larger N, although the natural variability is now larger than the previous example, leaving the second term dominating the uncertainty on the mean, which is

the arithmetical mean of the retrieval uncertainties. Since for cloud optical thickness the retrieval uncertainty is usually highest for clouds with high COTs, the uncertainty of the mean is highest in regions dominated by such clouds.

The Cloud_cci Level-3C products contain the individual uncertainty terms $\sigma_{\text{std}}$, $\langle \sigma_i \rangle$ and $\langle \sigma_i^2 \rangle$ for each variable in addition to the mean. This allows posterior calculations of $\sigma_{\langle x \rangle}$ and $\sigma_{\text{true}}$ for any given uncertainty correlation.

## 3  Product examples

In this section most Cloud_cci products are discussed using the example of the AVHRR-PM dataset, i.e. (1) Level-3U data of NOAA-18/AVHRR for 2008/06/22 and (2) Level-3C data for the month of June 2008. Figure 5 shows CMA, CFC, CPH, liquid cloud fraction, COT and CER. Figure 6 shows CTP, LWP, IWP and CLA. In Figure 7 the JCH is presented in two ways: (1)

shown as a global COT-CTP histogram aggregated over all grid cells and (2) the relative fraction of a certain subset of clouds, in this case cumulus clouds according to the ISCCP definition given in Rossow and Schiffer (1999): clouds with CTP larger than 680hPa and with COT lower than 3.6.

In Figure 8 time series of selected cloud properties are shown for monthly, latitude-weighted averages (within a 60°S-60°N latitude band) of Cloud_cci AVHRR-AM, MODIS-Terra, ATSR2-AATSR and MERIS+AATSR datasets. All datasets

are relatively stable. However, the time series exhibit some systematic offsets between the datasets. Though these offsets have not been investigated in detail yet, they appear to be related to differences in the spectral characteristics of the AVHRR heritage channels used, i.e. the position and shape of the spectral response functions. Also, differences in the spatial resolution of AVHRR (footprint size of 1 x 4km) compared with MODIS and AATSR (1km x 1km footprint size) may have a significant impact. Figure 9 shows the time series for Cloud_cci AVHRR-PM and MODIS-Aqua datasets. Considering the time series

of all datasets, some inhomogeneities are found for AVHRR-AM, AVHRR-PM. These are mainly due to differences in local observation time of the individual satellites. A significant portion of this is due to a growing delay in local observation time with satellite lifetime caused by the drift of the satellite orbit. For AVHRR-AM, there is an additional jump in local observation





**Figure 4.** Monthly standard deviation (a) and monthly mean (b) for cloud optical thickness (COT). Panels (c) and (d) show the estimated natural variability and uncertainty of the mean (d) for a correlation of 0.1. Panel (e) and (f) are as panels (c) and (d) but for an uncertainty correlation of 1.0. All data is from AVHRR-PM in 2008/06.

time between the early morning orbits of NOAA-12 and NOAA15, and the mid-morning orbits of NOAA-17 and Metop-A. Variability in local observation time means that different parts of a diurnal cycle of clouds are sampled. Also, the solar



zenith angle and the relative azimuth angle between satellite and sun change with local observation time and can also lead to inhomogeneities in a time series. Statistical correction methods for mentioned effects exist (e.g. Foster and Heidinger, 2013) and should be applied precedent to any trend analysis. The impact of spectral deviation among the individual AVHRR sensors of a dataset is assumed to have a smaller impact compared to the satellite drift effect. For ATSR2-AATSR a small jump in the time series of some cloud properties is found at the sensor transition. This primarily driven by differences in the dynamic range

of the 3.7μm channel. The channel saturates more often for ATSR-2. This is particularly evident for CFC, LWP and IWP.

Another feature in the AVHRR-PM CFC time series (Figure 9) is worth mentioning. In 1982 (and onwards) and in 1991 (and onwards) positive anomalies are found which are related to the major eruptions of the El Chichón (Mexico) and Pinatubo (Philippines) volcanoes. A first analysis revealed that heavy aerosol loadings are sometimes detected as clouds. As this seems to be a general feature of CC4CL and FAME-C, cloud detection and cloud-top properties of all datasets should be used with

caution in heavy aerosol conditions.





**Figure 5.** Level-3U (left panels) and Level-3C (right panels) of cloud mask/fraction (a-b), cloud phase/liquid cloud fraction (c-d), optical thickness (e-f) and effective radius (g-h) for the AVHRR-PM dataset for 2008/06. For the Level-3U examples, the ascending nodes of the orbits are shown, which roughly correspond to the daylight portions of the orbits of NOAA-18. COT, LWP, IWP and CLA are only retrieved during daytime conditions. Areas with no valid retrievals in this day/month are gray-shaded.





**Figure 6.** As Figure 5 but for cloud-top pressure (a-b), liquid water path (c-d), ice water path (e-f), spectral cloud albedo at 0.6μm (g-h).



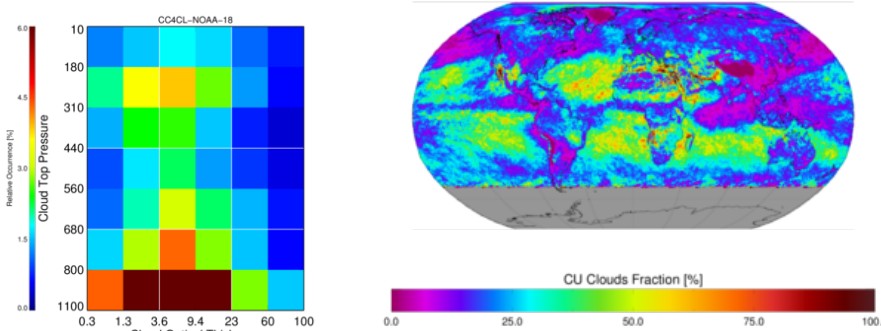

**Figure 7.** Left: Joint cloud property histogram, globally aggregated over all grid cells. Right: Global map of relative occurrence of cumulus clouds (according to ISCCP definition of Rossow and Schiffer (1999) with CTP > 680hPa and COT < 3.6) with respect to all clouds. Data shown is from AVHRR-PM datasets. Data shown is from AVHRR-PM in 2008/06.





**Figure 8.** Times series of monthly mean cloud properties of Cloud_cci datasets, with thin lines being (from top to bottom): Time series of monthly mean cloud fraction (CFC), liquid cloud fraction (CPH), cloud-top pressure (CTP), cloud optical thickness (COT), cloud effective radius (CER), liquid water path (LWP) and ice water path (IWP); overlaid with a running average (thick lines). Shown are those Cloud_cci datasets that are based on so-called morning satellites. The monthly means are calculated for 60°S-60°N (latitude-weighted). The time series shown have not been corrected for satellite drift or diurnal cycle.





**Figure 9.** As Figure 8 but for Cloud_cci datasets based on afternoon satellites. The time series shown have not been corrected for satellite drift or diurnal cycle.





## 4  Validation summary

Cloud_cci datasets between 2006 and 2014 were collocated in space and time with observations from the CALIOP instrument mounted on board the CALIPSO satellite. The active measurements of CALIOP observations can be considered as accurate reference for CMA, CPH and CTH. However, it is important to note that cloud properties from CALIOP are physically different to that given by the Cloud_cci products. For CTH for example the active sensor detects where the density of particles sharply increases - the physical cloud top edge. Passive sensors observe the entire atmospheric column simultaneously. The CTH retrieved from passive sensors is an effective average through the cloud top layer. As such, it is expected that Cloud_cci CTHs are lower than those observed by CALIOP, in particular for clouds containing fuzzy (semi-transparent) cloud layers. To account for this, various metrics have been considered for the validation of Cloud_cci cloud properties, where the CALIOP properties have been adjusted with respect to the optical depth profile.

The CAL_LID_L2_05kmCLay-Prov product has been used, which was downloaded from ICARE Data and Service Center (http://www.icare.univ-lille1.fr). All collocations are based on searching for the nearest-neighbour in the Cloud_cci Level-3U data to each CALIOP observation. Due to the similar orbital characteristics of NOAA-18, NOAA-19 (both are part of AVHRR-PM), Aqua and CALIPSO, a very large set of collocations between these passive imagers and CALIOP was found with only short temporal mismatches. A time window of $\pm 3$ minutes was used in in these cases. The orbital characteristics of NOAA-17, Metop-A (both are part of the AVHRR-AM dataset), Envisat (part of the ATSR2-AATSR and MERIS+AATSR datasets) and Terra deviate significantly from CALIPSO. Thus, for these satellites the collocation time window was extended to $\pm 15$ minutes. These collocations are however still limited to the very high latitudes around 70 degrees north and south, thus are occasionally affected by snow and ice as well as low solar-zenith-angle conditions.

In Table 6 the validation results for CMA are presented, i.e. the probabilities of correctly detecting cloudy and clear-sky scenes (Hitrate, $POD_{cloudy}$, $POD_{clear}$; see Appendix A for the definition of these terms), the bias and the number of considered pixels.. In a first set-up, the distinction of clear-sky and cloudy scenes in CALIOP data was made without applying any cloud optical depth threshold (upper part of Table 6). The hitrates (i.e. the fraction of pixels that were correctly labelled clear or cloudy by the Cloud_cci CMA wrt. CALIOP) range from 73 to 91% with the highest values for MODIS-Terra. The biases range from -13 to -1% indicating a slight underestimation of cloudiness, despite the fact that the $POD_{cloudy}$ are significantly higher than $POD_{clear}$ for all datasets, which can be explained by the higher frequency of cloudy scenes compared to clear-sky scenes. Removing optically very thin clouds from the CALIPO data (i.e. classifying all clouds with optical thickness lower than 0.15 as clear-sky in the CALIOP data) significantly improves the agreement of Cloud_cci data with CALIOP (lower part of Table 6). The hitrates are mostly increased (except for MODIS-Terra) and the biases are reduced (except MODIS-Terra and ATSR2-AATSR).

In Table 7 the validation results for CPH are presented, i.e. the probabilities of correctly detecting liquid and ice clouds (Hitrate, $POD_{liquid}$, $POD_{ice}$; see Appendix A for the definition of these terms), the bias and the number of considered cloudy pixels. As for CMA, two validation set-ups have been used: in the first the CALIOP cloud phase from the uppermost reported cloud layer was used (upper part of Table 7), in the second set-up the CALIOP cloud phase was taken at that level in the





cloud at which the top-down cloud optical depth is 0.15 or higher (COD$_{lev}$=0.15; lower part of Table 7). For the first set-up, the probabilities of detecting the correct phase range from 72 to 78% with highest values for AVHRR-PM and ATSR2-AATSR. All datasets except MERIS+AATSR and ATSR2-AATSR show a clear liquid-bias, meaning an overestimation of the occurrence of liquid clouds at the cost of ice clouds. A liquid-bias can be explained by a lack of sensitivity of the passive imager retrievals

to optically very thin ice cloud layers above liquid cloud layers. This is supported by the lower POD$_{ice}$ values compared to POD$_{liquid}$. The hitrates for phase agreement between Cloud_cci datasets and CALIOP increase for COD$_{lev}$=0.15, which is mainly driven by a better detection of ice phase (increased POD$_{ice}$) while at the same time the detection efficiency for liquid phase (POD$_{liq}$) decreases only slightly. The biases change towards ice (reduced liquid bias or ice bias instead of liquid bias). The results show that the correct cloud phase determination for passive imager retrieval is very sensitive to phase changes in

the uppermost cloud layers (e.g. between the physical cloud top and 1 optical depth into the cloud). In addition to the CPH comparisons presented, the scores were again calculated including only conditions for which no phase change occurred in the CALIOP data between the physical cloud top and 1 optical depth into the cloud (not shown). The agreement between Cloud_cci datasets and CALIOP data further improves significantly. Hitrates increase by 2 to 10%, mainly driven by a much better detection probability for ice clouds.

In Table 8 the validation results for CTH are presented, i.e. standard and mean deviations. The comparisons were limited to those collocated pixels for which both CALIOP and the Cloud_cci dataset report clouds and valid retrievals of cloud phase and cloud-top height. In addition, the data was restricted to cases were the phase assignment between CALIOP and the Cloud_cci dataset is in agreement. Removing this uncertainty in the phase assignment gives a clearer picture of the actual cloud-top height retrieval, since the first guess of COT, CER and CTP in the Cloud_cci retrieval systems is a function of the prior determined

cloud phase. The comparisons were separated into liquid and ice cloud conditions and carried out twice using different top-down cloud optical depth levels (COD$_{lev}$) at which the reference CTH was selected from CALIOP profiles. For liquid clouds the standard deviation between Cloud_cci datasets and CALIOP is around 1km and the bias is generally below 0.2km (except MERIS+AATSR with a bias of 0.79km). These values do not change significantly when selecting the reference CALIOP CTH at COD$_{lev}$=0.15 (bottom part of Table 8), which indicates that water clouds usually do not have small optical thicknesses.

For ice clouds the agreement to CALIOP is lower as expected. Standard deviations range from 1.9 to 2.8km and the bias is generally negative (thus an underestimation of CTH for ice clouds in Cloud_cci datasets) between -2.5 and -3.6km. These negative biases are reduced to -1.9 to -2.8km when the CALIOP CTH is taken at COD$_{lev}$=0.15. Standard deviations are also reduced by about 0.2km for this setting. The agreement of the Cloud_cci ice cloud CTH to CALIOP further improves with increasing COD$_{lev}$ (not shown). For example, at COD$_{lev}$=1.0 the biases for ice cloud CTHs are decreased to -0.78 to -1.99km.

LWP retrievals were validated against LWP data derived from satellite-based, passive microwave (MW) data (O'Dell et al., 2008). Microwave radiation can normally fully penetrate clouds.Thus, MW measurements can provide a direct measurement of the total liquid cloud condensate amount. Shortcomings of the MW data usually exist for scenes with low LWP and scenes with clouds that also contain large solid (ice) and liquid (rain) particles. Because of this and because of the different orbital characteristics of the MW-sensor carrying satellites our validation focused on Level-3C (i.e. monthly averages) in three stra-

tocumulus regions, for which ice cloud occurrence is very low. The regions are: the oceanic area west of Africa at 10°S-20°S,





0°-10°E (SAF hereafter), the oceanic area west of South America at 16°S-26°S, 76°W-86°W (SAM hereafter), and the oceanic area west of California at 20°N-30°N, 120°-130°W (NAM hereafter). The O'Dell et al. (2008) data has an accuracy of 15-30% and contains monthly mean diurnal cycle products, from which the 10:30 AM and 01:30 PM values were taken to match the Cloud_cci morning and afternoon sensors, respectively. For the validation scores presented in Table 9, only the common over-

lap period among all Cloud_cci datasets and the MW data was considered (2003 to 2008). The validation scores vary among the Cloud_cci datasets but also among the three regions under consideration. Considering the correlation coefficients, the SAF region exhibits the best agreement to the MW for all Cloud_cci datasets, which might be due to the large seasonal cycle of LWP in this region. Bias and bias-corrected root mean square errors (bc-RMSEs) do not differ from the other regions, except for the afternoon satellite datasets AVHRR-PM and MODIS-Aqua, which show best scores for SAF. The ATSR2-AATSR and

MERIS+AATSR datasets have the largest deviations to the MW data for all three regions. For all other datasets very small positive or moderately negative biases are found, thus a slight underestimation of LWP compared to MW. Considering the given uncertainty estimates for the reference data (15-30%), one can still conclude that there is an agreement between all Cloud_cci datasets and MW in nearly all regions. It is worth mentioning that the agreement to MW reduces when considering the time period before 2003 for AVHRR-AM, AVHRR-PM and ATSR2-AATSR. This is mainly due to problems for the earlier

satellites, which can also be seen in the LWP time series plots of Figures 8 and 9.

Beyond the limited validation results presented in this paper, a comprehensive effort has been carried out to compare the Cloud_cci datasets with other, well-established datasets such as PATMOS-x, CLARA-A2 and MODIS Collection 6 (Stapelberg et al., 2017). Their results prove the quality of the Cloud_cci datasets.

**Table 6.** Summary of cloud mask (CMA) validation results for Cloud_cci datasets when compared against CALIOP. Validation measures are the probabilities of detecting cloudy and clear scenes (Hitrate, $POD_{liquid}$, $POD_{ice}$; see Appendix A for the definition of these terms) and bias. Also given is the number of collocated pixels. The scores are separated into two cloud optical thickness thresholds ($COT_{thres}$) representing above which CALIOP COT the CALIOP pixel was classified cloudy.

| | Score | AVHRR-AM | AVHRR-PM | MODIS-Terra* | MODIS-Aqua | ATSR2-AATSR* | MERIS+AATSR* |
|---|---|---|---|---|---|---|---|
| $COT_{thres} = 0.0$ | Hitrate [%] | 76.2 | 81.2 | 91.0 | 81.6 | 74.3 | 73.4 |
| | $POD_{cloudy}$ [%] | 78.3 | 79.0 | 94.4 | 81.1 | 80.0 | 74.6 |
| | $POD_{clear}$ [%] | 70.4 | 87.7 | 70.0 | 83.1 | 60.5 | 69.6 |
| | Bias [%] | -8.1 | -12.6 | -1.1 | -9.8 | -2.6 | -12.5 |
| | Number | 42 119 | 16 675 575 | 19 118 | 16 494 437 | 94 039 | 23 098 |
| $COT_{thres} = 0.15$ | Hitrate [%] | 78.3 | 84.9 | 89.4 | 83.5 | 75.0 | 75.6 |
| | $POD_{cloudy}$ [%] | 84.4 | 87.5 | 96.5 | 88.7 | 86.2 | 80.3 |
| | $POD_{clear}$ [%] | 67.6 | 80.5 | 51.7 | 75.0 | 58.4 | 66.4 |
| | Bias [%] | 1.9 | -0.5 | 4.8 | 2.3 | 8.5 | -1.5 |
| | Number | 42 119 | 16 675 575 | 19 118 | 16 494 437 | 94 039 | 23 098 |

* Time window used for collocations was ±15 minutes for ATSR2+AATSR, MERIS+AATSR and MODIS-Terra. For all others a ±3 minutes window was used.





**Table 7.** Summary of cloud phase (CPH) validation results for Cloud_cci datasets when compared against CALIOP. Validation measures are probability of detection (POD) and bias of liquid cloud occurrence. Also given is the number of collocated pixels. The scores are separated into two cloud optical depth levels ($COD_{lev}$) representing at which top-down COD into the cloud the reference CALIOP CPH was taken.

| | Score | AVHRR-AM | AVHRR-PM | MODIS-Terra* | MODIS-Aqua | ATSR2-AATSR* | MERIS+AATSR* |
|---|---|---|---|---|---|---|---|
| $COD_{lev} = 0.0$ | Hitrate [%] | 72.0 | 77.1 | 74.4 | 73.7 | 77.6 | 72.3 |
| | $POD_{liq}$ [%] | 71.3 | 78.0 | 88.1 | 82.7 | 67.7 | 67.3 |
| | $POD_{ice}$ [%] | 72.5 | 76.5 | 65.1 | 68.1 | 83.1 | 76.9 |
| | Bias [%] | 6.4 | 5.9 | 15.9 | 13.0 | -0.6 | -3.5 |
| | Number | 23 151 | 9 436 914 | 15 581 | 9 576 169 | 50 659 | 9 210 |
| $COD_{lev} = 0.15$ | Hitrate [%] | 76.0 | 80.6 | 80.7 | 79.6 | 77.8 | 73.8 |
| | $POD_{liq}$ [%] | 70.8 | 74.0 | 85.7 | 79.6 | 64.6 | 64.2 |
| | $POD_{ice}$ [%] | 81.2 | 87.4 | 74.6 | 79.6 | 89.7 | 88.6 |
| | Bias [%] | -5.1 | -6.9 | 5.0 | -0.27 | -11.3 | -17.3 |
| | Number | 22 221 | 8 935 688 | 15 213 | 8 951 056 | 47 527 | 8 770 |

\* Time window used for collocations was ±15 minutes for ATSR2+AATSR, MERIS+AATSR and MODIS-Terra. For all others a ±3 minutes window was used.

**Table 8.** Summary of cloud-top height (CTH) validation results for Cloud_cci datasets when compared against CALIOP. Validation measures are standard deviation (Std) and bias. Also given is the number of collocated pixels. All scores are separated into liquid and ice clouds (both Cloud_cci dataset and CALIOP had to agree on phase) and into two cloud optical depth levels ($COD_{lev}$) representing at which top-down COD into the cloud the reference CALIOP CTH was taken.

| | Score | AVHRR-AM | AVHRR-PM | MODIS-Terra* | MODIS-Aqua | ATSR2-AATSR* | MERIS+AATSR* |
|---|---|---|---|---|---|---|---|
| $COD_{lev} = 0.0$ | $Std_{liq}$ [km] | 1.04 | 0.91 | 0.75 | 0.97 | 0.89 | 1.35 |
| | $Bias_{liq}$ [km] | 0.17 | -0.12 | 0.04 | -0.09 | 0.11 | 0.79 |
| | $Number_{liq}$ | 6 177 | 2 850 732 | 5 552 | 4 041 688 | 12 149 | 2 836 |
| | $Std_{ice}$ [km] | 2.66 | 2.84 | 1.91 | 2.79 | 2.33 | 1.90 |
| | $Bias_{ice}$ [km] | -2.66 | -2.65 | -2.65 | -2.57 | -2.59 | -3.59 |
| | $Number_{ice}$ | 10 468 | 4 417 179 | 6 041 | 4 014 001 | 27 164 | 3 614 |
| $COD_{lev} = 0.15$ | $Std_{liq}$ [km] | 1.06 | 0.97 | 0.85 | 1.03 | 0.94 | 1.35 |
| | $Bias_{liq}$ [km] | 0.14 | -0.09 | 0.08 | -0.06 | 0.13 | 0.82 |
| | $Number_{liq}$ | 7 807 | 3 335 953 | 6 688 | 3 596 738 | 14 508 | 3 300 |
| | $Std_{ice}$ [km] | 2.56 | 2.59 | 1.73 | 2.51 | 2.04 | 1.79 |
| | $Bias_{ice}$ [km] | -1.96 | -1.94 | -2.21 | -1.90 | -1.95 | -2.82 |
| | $Number_{ice}$ | 9 046 | 3 865 027 | 5 521 | 3 524 716 | 22 455 | 3 001 |

\* Time window used for collocations was ±15 minutes for ATSR2+AATSR, MERIS+AATSR and MODIS-Terra. For all others a ±3 minutes window was used.



**Table 9.** Summary of liquid water path (LWP) validation results for Cloud_cci datasets when compared against the passive, microwave-based (MW), monthly mean LWP data of O'Dell et al. (2008) in the period 2003 to 2008. The validation was performed for three oceanic stratocumulus regions for which the frequency of ice cloud occurrence is very small, SAF: west of Africa (10°S-20°S, 0°-10°E); SAM: west of South America (16S-26S, 76W-86W); NAM: west of California (20°N-30°N, 120°W-130°W) - see text for details. Reported are the mean LWP of the MW as well as the bias and standard deviation (Std) for each Cloud_cci dataset with respect to the MW (all values in $g/m^2$). In addition, the values are given in relative terms in % in brackets. The correlation coefficient (r) is also given.

| Region | Score | AVHRR-AM | AVHRR-PM | MODIS-Terra | MODIS-Aqua | ATSR2-AATSR | MERIS+AATSR |
|--------|-------|----------|----------|-------------|------------|-------------|-------------|
| NAM | Mean* | 60.1 | 46.2 | 60.1 | 46.2 | 60.1 | 60.1 |
| | Bias | -6.8 (-11%) | -6.7 (-14%) | -14.1 (-23%) | -1.8 (-3%) | 5.7 (9%) | 0.6 (1%) |
| | Std | 5.4 (8%) | 8.2 (17%) | 5.4 (8%) | 7.0 (15%) | 13.4 (22%) | 11.3 (18%) |
| | r | 0.88 | 0.66 | 0.88 | 0.75 | 0.80 | 0.72 |
| SAM | Mean* | 80.2 | 52.6 | 80.2 | 52.6 | 80.2 | 80.2 |
| | Bias | -1.2 (-1%) | -3.9 (-7%) | -8.5 (-10%) | 0.7 (1%) | 27.5 (34%) | 13.2 (16%) |
| | Std | 7.4 (9%) | 7.1 (13%) | 6.8 (8%) | 7.6 (14%) | 13.2 (16%) | 10.3 (12%) |
| | r | 0.88 | 0.93 | 0.90 | 0.93 | 0.83 | 0.72 |
| SAF | Mean* | 59.5 | 39.6 | 59.5 | 39.6 | 59.5 | 59.5 |
| | Bias | -7.6 (-12%) | -2.4 (-6%) | -11.9 (-19%) | 1.0 (2%) | 11.8 (19%) | 1.2 (2%) |
| | Std | 9.7 (16%) | 5.4 (13%) | 6.3 (10%) | 5.3 (13%) | 11.0 (18%) | 10.7 (17%) |
| | r | 0.89 | 0.95 | 0.96 | 0.95 | 0.93 | 0.85 |

* The mean values given are for the reference data at 10:30 AM when compared against AVHRR-AM, MODIS-Terra, ATSR2-AATSR and MERIS+AATSR, and at 01:30 PM when compared against AVHRR-PM and MODIS-Aqua.

## 5   Summary

In this paper cloud property datasets generated within the ESA Cloud_cci project were presented. The datasets are based on passive imager measurements on board polar-orbiting satellites. The measurement records have been characterized carefully and, in case of AVHRR, been re-calibrated. Two retrieval systems were developed: (1) the Community Cloud retrieval for CLimate (CC4CL) which was applied to AVHRR as well as to the AVHRR heritage channels measured by MODIS, ATSR2 and AATSR, and (2) the Freie Universität Berlin AATSR MERIS Cloud (FAME-C) which was applied to combined MERIS+AATSR measurements.

Based on these new developments, global cloud climatologies were generated for all mentioned sensors spanning their entire life time. The datasets are named: AVHRR-AM, AVHRR-PM, MODIS-Terra, MODIS-Aqua, ATSR2-AATSR and MERIS+AATSR. The cloud properties derived are: cloud mask/fraction, cloud phase, cloud-top pressure/height/temperature, cloud optical thickness, cloud effective radius, liquid/ice water path and spectral cloud albedo. The data is available as pixel-based retrievals (Level-2), globally gridded composites (Level-3U) and monthly summaries of the cloud properties (Level-3C): averages, standard deviations and histograms. The OE-based uncertainty information per pixel (contained in Level-2 and Level-3U), was propagated into Level-3C data using an introduced mathematical framework (Equations 4 and 5). While the main



characteristics of all datasets are very similar to the AVHRR-PM examples shown, it needs to be noted that some deviations exist. These are mainly introduced by differences in spatio-temporal observation frequency, remaining differences in spectral properties among the considered sensors and differences in retrieval systems, i.e. for MERIS+AATSR dataset.

Level-2 validation of cloud mask, cloud phase and cloud-top height against CALIOP revealed a probability of correct detection of cloudy and clear-sky scenes between 70% and 90% (hitrates), with a general underestimation of cloud occurrence frequency when compared to all detected clouds in CALIOP data, which reflects the detection limitation of passive imagers. Neglecting optically very thin clouds in CALIOP data improves the agreement in terms of probability of detection and biases. For cloud phase, hitrates of 70-80% are reached with a bias towards liquid clouds (except the MERIS+AATSR dataset) when comparing to the uppermost cloud layer of CALIOP data. When comparing against the CALIOP phase taken at a top-down cloud optical depth of 0.15 into the cloud, hitrates increase by approximately 5% along with a reduction in the biases. Validating cloud-top height gives generally small standard deviations and biases for liquid clouds, while the agreement to CALIOP is lower for ice clouds. For ice clouds, a strong dependence on the reference level from which the CALIOP cloud-top height is taken is found. Biases reduce to -0.8 to -1.99 km when the reference CALIOP cloud-top height is taken at a top-down optical depth of 1 into the cloud-top. Monthly mean liquid water path was validated against passive, microwave-based satellite data. The mean and standard deviations are relatively diverse and strongly dependent on the dataset and region. However, for most Cloud_cci datasets and considered regions an agreement to the reference data within the reported uncertainty of the reference data (15-30 %) was found.

The validation results presented here, as well as the very comprehensive Cloud_cci validation report (Stapelberg et al., 2017) have proven the comparability of Cloud_cci datasets with already existing datasets of the same kind. However, additionally ensuring spectral consistency and adding rigorously propagated uncertainty measures make the Cloud_cci datasets distinct from them. The evaluation process of the presented datasets has also revealed some limitations, of which the most important ones are listed in Appendix B. In the near future, the Cloud_cci retrieval systems will undergo a revision, e.g. improving the forward models and LUTs and revising the BRDF for snow and ice surface). Based on these developments the datasets will be reprocessed, also including more recent time periods. Along with that, the product portfolio will be extended to include broad-band radiation flux properties at surface and TOA, which will allow several new applications such as studying the cloud radiative effect.

## 6 Data availability

All presented Cloud_cci datasets are freely available. DOIs have been issued for all datasets (see Table 1) with each DOI landing page containing a brief summary of the corresponding dataset and a link to the data access page (http://www.esa-cloud-cci.org/?q=data_download).



## Appendix A: Scores

The calculation of the probabilities of detection (PODs) and the hitrates of binary events (e.g. clear-sky/cloudy, liquid/ice phase) is based on the entries of the contingency table (Table A1).

**Table A1.** Contingency table.

|  | Reference = 0 | Reference = 1 |
|---|---|---|
| Cloud_cci = 0 | $N_{00}$ | $N_{01}$ |
| Cloud_cci = 1 | $N_{01}$ | $N_{11}$ |

The POD for a certain event (e.g. event 0) is determined according to Equation A1. Thus, it is defined as the number of the agreements with the reference for a certain event, divided by the total number of this event in the reference data.

$$\mathrm{POD}_0 = \frac{N_{00}}{(N_{00} + N_{10})}. \tag{A1}$$

The calculation of the Hitrate is given in Equation A2. The hitrate is defined as the number of cases in which an agreement with the reference data was found, divided by the number of all cases.

$$\mathrm{Hitrate}_0 = \frac{(N_{00} + N_{11})}{(N_{01} + N_{10} + N_{00} + N_{11})}. \tag{A2}$$

## Appendix B: Known limitations

The most significant limitations of the Cloud_cci datasets, as revealed during the evaluation, are listed. As not all limitations apply to all datasets, a mapping table is provided (Table B1).

(1) Cloud detection shortcomings (overestimating of CMA and CFC) in conditions of high aerosol loadings, e.g. severe volcano eruptions or (local) heavy desert dust outbreaks.

(2) Cloud detection shortcomings during polar night due to missing visible information and very cold surface temperatures (mainly affecting CMA and CFC).

(3) Shortcomings in cloud detection (affecting CMA and CFC) and optical property retrievals (CER, COT, LWP, IWP, CLA) in regions with high surface reflection of solar radiation, e.g. in areas of sun glint over ocean or in land areas with snow, ice and desert soil surfaces (high albedo).

(4) Instabilities in the time series (of all cloud variables) due to satellite drift and/or switch in local overpass time. Satellite drift or diurnal cycle correction is required before using the datasets for trend analysis.

(5) Instabilities in the time series due to switch of NIR channels (affecting mainly the retrieval of CER and thus LWP and IWP): (a) during 2 years of NOAA-16 the AVHRR 1.6μm channel is switched on during daytime, while for the rest of the





AVHRR-PM time series the 3.7μm channel is measured during daytime, (b) in the AVHRR-AM time series, NOAA-12 and NOAA-15 have the 3.7μm channel measuring during daytime while NOAA-17 and Metop-A have 1.6μm .

(6) Significant overestimation of CER of ice clouds and IWP due to erroneous composition of radiative transfer look-up tables.

(7) Overestimation of CER and COT for snow/ice surfaces and high solar zenith angles

(8) The AVHRR on NOAA-12 and NOAA-15 satellites measure in near-twilight conditions, due to the early morning orbits of these two satellites, for which the retrieval of all cloud properties, especially the optical properties, are very difficult.

(9) Inconsistencies in the 3.7μm channel between the ATSR-2 and AATSR affected CPH, CMA and CFC.

(10) Additional errors introduced when converting cloud-top level properties (CTH, CTP and CTT) to each other using potentially incorrect model profiles. However, these errors are assumed to be significantly smaller than the actual retrieval errors of CTP.

(11) Larger errors in cloud property retrieval (all properties except CMA and CFC) in multi-layer cloud conditions in particular when a high, optically thin ice cloud overlays a optically thick, lower liquid cloud. See McGarragh et al. (2017a) for an attempt to capture cloud properties from multiple cloud layers.

(12) Larger sampling error (affecting all cloud properties) accompanied by artificially increased observed variability due to low observation frequency for some sensors.

**Table B1.** Mapping the listed limitations above to the Cloud_cci datasets they apply to.

| AVHRR-AM | AVHRR-PM | MODIS-Terra* | MODIS-Aqua | ATSR2-AATSR* | MERIS+AATSR* |
|---|---|---|---|---|---|
| 1,2,3,4,5a,6,7,8,10,11 | 1,2,3,4,5b,6,7,10,11 | 1,2,3,6,7,10,11 | 1,2,3,6,7,10,11 | 1,2,3,6,7,9,10,11,12 | 1,3,7,10,11,12 |

*Competing interests.* The authors declare that no competing interests are present.

*Acknowledgements.* This work was supported by the European Space Agency (ESA) through the Cloud_cci project (contract No.: 4000109870/13/I-NB). The authors would like to acknowledge the help of NASA Goddard Space Flight Center in providing MODIS Collection 6 Level-1 data and the help of NOAA and the University of Wisconsin for providing the AVHRR GAC Level-1 data and corresponding intercalibration coefficients for the visible and near-infrared channels of AVHRR.



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
