# Peer review of "Cloud property datasets retrieved from AVHRR, MODIS, AATSR and MERIS in the framework of the Cloud\_cci project"

_Earth System Science Data, 2017_

## Referee Comment (RC1) · Anonymous Referee #1 · 21 Jul 2017

Review comments on manuscript "Cloud property datasets retrieved from AVHRR, MODIS, AATSR and MERIS in the framework of the Cloud_cci project"

Author(s): Martin Stengel et al.

MS No.: essd-2017-48

General comments:

This paper presents the cloud property dataset as a result of the Cloud_cci project. The dataset includes products such as cloud optical thickness, effective radius, thermodynamic phase, cloud top pressure/height/temperature, liquid/ice water path and spectral

albedo, which are retrieved using AVHRR, MODIS, ATSR2, AATSR, and MERIS observations with the optimal estimation technique. This unified dataset will be helpful for long term cloud/climate related studies. The materials are presented well. I recommend publication of the paper after some revisions listed below.

One general concern I have is about the justification for generating this dataset. For each of the sensors/mission used in the paper, there exist corresponding mature retrieval datasets, which are validated and widely used by the community. One can understand the new dataset presented by the paper applied self-consistent algorithms to all the sensors, which is definitely a desirable thing to do, but the ultimate goal is to have a dataset with higher quality than the existing ones. I would suggest the authors to include results or at least a further discussion on why the new dataset is better than a congregation of each of the individual dataset. This could be achieved through showing that either the quality of the new dataset is better or the new dataset can be used for work that cannot be done otherwise (may be due to the inconsistency of algorithms).

Specific comments:

1) P1, Line 15 in Abstract: "though" suppose to be "through"

2) P3: "developing physical retrieval systems for cloud properties with spectral consistency over all utilized bands". Take the 36-channel MODIS as an example, how many bands are utilized and how is the spectral consistency achieved?

3) P5, Line 5 in Section 2.1: "night-time node)" unmatched ")"

4) P15: regarding the offsets between the datasets shown in Fig 8. It's puzzling to me why "they appear to be related to differences in the spectral characteristics of AVHRR heritage channels", as the spectral response functions should have been applied in the radiative transfer calculations. Could it be a calibration issue?

5) P19: Fig. 6g: should be "cloud ice water path"; the word "ice" is missing in the color bar.

6) P28, Line 23: "surface)", unmatched ")"

---

## Referee Comment (RC2) · Anonymous Referee #2 · 29 Aug 2017

This manuscript reports new cloud property datasets Cloud_cci products based on measurements from the passive imaging satellite sensors AVHRR, MODIS, ATSR2, AATSR and MERIS. The authors developed two retrieval systems that include components for cloud detection and cloud typing followed by cloud property retrievals based on the optimal estimation (OE) technique, to simultaneously retrieve cloud-top pressure, cloud particle effective radius and cloud optical thickness using measurements at 5 visible, near-infrared and thermal infrared wavelengths, which ensures spectral consistency. The retrieved cloud properties are further processed to derive cloud-top height, cloud-top temperature, cloud liquid water path, cloud ice water path and spectral cloud albedo. These developments are useful for cloud and climate studies. The

spectral consistency of retrievals is especially important for deriving real physical properties of clouds from satellite data. These datasets can be useful for the research communities, so this reviewer recommends this manuscript be published. However, the presentation quality of this manuscript has not met the journal requirement and thus need significant improvement.

1. The presentation language is not clear and simple as required commonly by scientific papers, such as "Satellite-based datasets of geophysical variables evolve periodically by revisiting and maturing two essential aspects: the underlying radiance records and the retrieval systems applied. The first aspect is usually addressed by tracking down and collecting historic and new satellite recordings, by characterizing the accuracy and stability of the full measurement record, and by applying new inter-calibration backwards through the entire record whenever new satellite missions provide better references. The second aspect is facilitated by: (1) continuously growing computer capabilities enabling the application of more advanced (and more computationally expensive) retrieval systems and the utilization of additional or more frequent auxiliary data which often undergo regular updates themselves, and (2) retrieval improvements which are often triggered when new satellite missions offer more accurate reference observations against which the retrieval systems can be validated." Is this the straightforward style of scientific presentation?

2. Unnecessary statements exist across the text, such as "AVHRR is a passive imaging sensor, where the source of measured radiation is not emitted by the instrument. Instead, the upwards reflected solar and emitted thermal radiation is measured at the top of the atmosphere (TOA). This is done in abutting pixels that assemble a seamless image." Are these not known by every scientist in this research field?

3. Abstract is not well written, must be seriously revised to make "what's new" very outstanding.

4. The whole manuscript is not well organized. It looks like a loose technical report,
but not a scientific paper.

---

## Author Comment (AC1) · 26 Sep 2017

Authors' response to Anonymous Referee #1

*Referee comment:*
One general concern I have is about the justification for generating this dataset. For each of the sensors/mission used in the paper, there exist corresponding mature retrieval datasets, which are validated and widely used by the community. One can understand the new dataset presented by the paper applied self-consistent algorithms to all the sensors, which is definitely a desirable thing to do, but the ultimate goal is to have a dataset with higher quality than the existing ones. I would suggest the authors to include results or at least a further discussion on why the new dataset is better than a congregation of each of the individual dataset. This could be achieved through showing that either the quality of the new dataset is better or the new dataset can be used for work that cannot be done otherwise (may be due to the inconsistency of algorithms).

*Author's response:*
We believe that already all individual datasets presented have their novel characteristics, either by being the first of its kind (e.g. MERS+AATSR: utilizing measurements of both sensors simultaneously.) or by showing better quality that precursor datasets (e.g. GRAPE ATSR2-AATSR) or generally by being based on a OE which ensures spectral consistency and which enables mathematically sound uncertainty propagation into the Level-2 cloud properties. In addition, all datasets feature Level-3 uncertainties that are based on a mathematically sound propagation of Level-2 uncertainties into Level-3 products.

Furthermore, a potential combination of all CC4CL-based sets (all except MERIS-AATSR) facilitates an increased temporal sampling thus a reduction of sampling errors compared to individual polar-orbiting sensors. In addition, studies of the impact of different temporal and spatial sampling as well as of spatial resolution of cloud climatologies are now feasible. Proving that the combination of the individual datasets leads to a dataset of higher quality will be subject to future studies, although this is a difficult task since reference measurements with high temporal resolution are necessary for this.

We rephrased the last two sentences of the manuscript to make this point clearer. We also added a paragraph to the summary section discussing the combinability of the datasets and new studies that are now possible.

*Author's changes to the manuscript:*
Last sentence of the abstract was modified to:
"In particular the ensured spectral consistency and the rigorous uncertainty propagation through all processing levels can be considered as new features of the Cloud_cci datasets compared to existing datasets. In addition, the consistency among the individual datasets allows for a potential combination of them as well as facilitates studies on the impact of temporal sampling and spatial resolution on cloud climatologies."

In addition we added the following discussion paragraph to the summary section:
"Applying the same retrieval system to multiple sensors also facilitates a combination of the individual datasets, ideally leading to higher quality. This will be subject to future studies. In addition to such a combination of the datasets, the consistency (i.e. by using the same retrieval system) among them will also enable studies that investigate the impact of the imaging characteristics of the different sensors on derived cloud climatologies. These imaging characteristics are for example the spatial resolution (1km x 1km for AATSR/MODIS vs. 1km x 4 km for AVHRR GAC) as well as the observation frequency driven by the sensor swath width (2399km for AVHRR vs. 500km for AATSR)."

**Specific comments:**

*Referee comment:*
1) P1, Line 15 in Abstract: "though" suppose to be "through"
*Author's response:*
Thanks. Has been corrected.

*Author's changes to the manuscript:*
Has been corrected.

*Referee comment:*
2) P3: "developing physical retrieval systems for cloud properties with spectral consistency over all utilized bands". Take the 36-channel MODIS as an example, how many bands are utilized and how is the spectral consistency achieved?
*Author's response:*
Information on both is already given in the manuscript, but it seems to be useful to point to the information behind the sentence mentioned by the referee.
*Author's changes to the manuscript:*
We added "(See above for definition of spectral consistency and see Section 2.1 for the set of the spectral bands that have been utilized for each sensor considered)" below the sentence indicated by the reviewer.

*Referee comment:*
3) P5, Line 5 in Section 2.1: "night-time node)" unmatched ")"
*Author's response:*
Has been corrected.
*Author's changes to the manuscript:*
Has been corrected.

*Referee comment:*
4) P15: regarding the offsets between the datasets shown in Fig 8. It's puzzling to me why "they appear to be related to differences in the spectral characteristics of AVHRR heritage channels", as the spectral response functions should have been applied in the radiative transfer calculations. Could it be a calibration issue?
*Author's response:*
Yes, it could also be a calibration issue. Our comment was more referring to the cloud detection and cloud typing for which no radiative transfer modelling (with sensor specific spectral response functions) was done. The cloud mask and typing procedure was developed for NOAA-18 AVHRR. We developed empirical adjustments to the measurements of MODIS and ATSR2/AATSR to mimic NOAA-18/AVHRR, but we assume these adjustments to be imperfect. For all AVHRR sensors other than NOAA18/AVHRR even no adjustment was done, even though the spectral characteristics among them differ as well. The imperfect mimic of NOAA-18/AVHRR probably causes systematic errors in cloud detection and cloud typing which will most likely also introduce systematic errors in the other variables either by including more or less cloudy pixels in the monthly means (shown) or by different retrieval setups depending on the phase selection. We will elaborate more on this in the manuscript. As suggested by the reviewer, we will also mention that imperfect calibration might have an impact too.
*Author's changes to the manuscript:*
We have changed the corresponding sentence to:
"Though these offsets have not been investigated in detail yet, it is currently assumed that they are caused by to the following three reasons. (1) Differences in the spectral characteristics of the AVHRR heritage channels used, i.e. the position and shape of the spectral response functions, which is only accounted for empirically in cloud detection and cloud typing schemes. (2) The applied, but maybe still imperfect, calibration of the measurement records. (3) Differences in the spatial resolution of AVHRR (footprint size of 1km x 4km) compared with MODIS and AATSR (1km x 1km footprint size) may have a significant impact."

*Referee comment:*
5) P19: Fig. 6g: should be "cloud ice water path"; the word "ice" is missing in the color bar.
*Author's response:*

We assume Fig. 6e is meant. Actually, both Fig. 6c and 6e (which are identical) show the cloud water path example, which is in Level-3U (daily, global composites) liquid water path in pixels identified as liquid clouds and ice water path in pixels identified as ice.

*Author's changes to the manuscript:*
We added the following sentence to the figure 6 caption: "…for Level-3U (left panels) and Level-3C (right panels) products. Panels (c) and (e) both show the Level-3U cloud water path, which represents liquid water path in liquid cloud pixels and ice water path in ice cloud pixels."

*Referee comment:*
6) P28, Line 23: "surface)", unmatched ")"
*Author's response:*
Has been corrected.
*Author's changes to the manuscript:*
")" was deleted. We also modified surface -> surfaces.

---

## Author Comment (AC2) · 26 Sep 2017

Authors' response to Anonymous Referee #2

*Referee comment:*
1. The presentation language is not clear and simple as required commonly by scientific papers, such as "Satellite-based datasets of geophysical variables evolve periodically by revisiting and maturing two essential aspects: the underlying radiance records and the retrieval systems applied. The first aspect is usually addressed by tracking down and collecting historic and new satellite recordings, by characterizing the accuracy and stability of the full measurement record, and by applying new inter-calibration backwards through the entire record whenever new satellite missions provide better references. The second aspect is facilitated by: (1) continuously growing computer capabilities enabling the application of more advanced (and more computationally expensive) retrieval systems and the utilization of additional or more frequent auxiliary data which often undergo regular updates themselves, and (2) retrieval improvements which are often triggered when new satellite missions offer more accurate reference observations against which the retrieval systems can be validated." Is this the straightforward style of scientific presentation?

*Author's response:*
Thank you for pointing this out. We have shortened and simplified this paragraph.

*Author's changes to the manuscript:*
The paragraph now reads: "Satellite-based datasets of geophysical variables are crucial for climate research as they represent observations of the Earth's climate system, which can be used for both the analysis of the climate and its variability as well as guidance for atmospheric model developments. These datasets evolve periodically by mainly two activities: (1) Extending and improving the underlying radiance record by adding new satellite recordings and applying new inter-calibration to the entire record. (2) The development and application of more advanced retrieval systems and the utilization of additional or more frequent auxiliary data which often undergo regular updates themselves."

*Referee comment:*
2. Unnecessary statements exist across the text, such as "AVHRR is a passive imaging sensor, where the source of measured radiation is not emitted by the instrument. Instead, the upwards reflected solar and emitted thermal radiation is measured at the top of the atmosphere (TOA). This is done in abutting pixels that assemble a seamless image." Are these not known by every scientist in this research field?

*Author's response:*
We agree, these statements are known to most experts in the remote sensing field. However, this paper presents the datasets that will also be used by users from other research fields and we find it meaningful to provide some basic information about the sensors that have been used to generate the datasets presented as this information is crucial for understanding all strengths and weaknesses.

*Author's changes to the manuscript:*
No changes.

*Referee comment:*
3. Abstract is not well written, must be seriously revised to make "what's new" very outstanding.

*Author's response:*
Most important aspects were already covered in the last sentence of the abstract. However, we revised this to make the new things clearer.

*Author's changes to the manuscript:*
Last sentence of the abstract was modified to: "In particular the ensured spectral consistency and the rigorous uncertainty propagation through all processing levels can be considered as new features of the Cloud_cci datasets compared to existing datasets. In addition, the consistency among the individual datasets allows for a potential combination of them as well as facilitates studies on the impact of temporal sampling and spatial resolution on cloud climatologies."

*Referee comment:*
4. The whole manuscript is not well organized. It looks like a loose technical report, but not a scientific paper.

*Author's response:*
We agree that our manuscript contains many scientific and technical details. In particular by including the latter we cannot entirely avoid similarities to a technical report. However, we believe that the content of our manuscript, and the way it is presented is in line with ESSD policy and typical ESSD papers.

*Author's changes to the manuscript:*
No changes.